# A Review of Research Progress in Selective Laser Melting (SLM)

**DOI:** 10.3390/mi14010057

**Published:** 2022-12-25

**Authors:** Bingwei Gao, Hongjian Zhao, Liqing Peng, Zhixin Sun

**Affiliations:** School of Mechanical and Power Engineering, Harbin University of Science and Technology, Harbin 150080, China

**Keywords:** 3D printing technology, selective laser melting, processing parameter, alloy, pottery and porcelain

## Abstract

SLM (Selective Laser Melting) is a unique additive manufacturing technology which plays an irreplaceable role in the modern industrial revolution. 3D printers can directly process metal powder quickly to obtain the necessary parts faster. Shortly, it will be possible to manufacture products at unparalleled speeds. Advanced manufacturing technology is used to produce durable and efficient parts with different metals that have good metal structure performance and excellent metal thermal performance, to lead the way for laser powder printing technology. Traditional creative ways are usually limited by time, and cannot respond to customers’ needs fast enough; for some parts with high precision and complexity, conventional manufacturing methods are inadequate. Contrary to this, SLM technology offers some advantages, such as requiring no molds this decreases production time and helps to reduce costs. In addition, SLM technology has strong comprehensive functions, which can reduce assembly time and improve material utilization. Parts with complex structures, such as cavities and three-dimensional grids, can be made without restricting the shape of products. Products or parts can be printed quickly without the use of expensive production equipment. The product quality is better, and the mechanical load performance is comparable to traditional production technologies (such as forging). This paper introduces in detail the process parameters that affect SLM technology and how they affect SLM, commonly used metal materials and non-metallic materials, and summarizes the current research. Finally, the problems faced by SLM are prospected.

## 1. Introduction

SLM was first proposed by the Fraunhofer Institute for Laser Technology in Germany, which was born in the 1990s, as a kind of 3D printing technology, a kind of technology that uses powder to melt under the heat of a laser beam, is then cooled, and solidifies to form. This makes it possible to arrange the crystal structures of various materials, which inspires higher-density parts that are almost only part designs and have better grade strength and robustness [1,2]. As a result of the melting of powder raw materials, the products produced have excellent surface smoothness. It can be used to directly form metal parts with nearly complete density. SLM technology has overcome the difficulties of complicated manufacturing processes of metal parts. The processing flow of SLM technology is shown in Figure 1.

Compared with the traditional metal processing technology, SLM technology can complete the manufacturing of parts without using molds. Therefore, the SLM process requires relatively less preparation, and has natural advantages in overall manufacturing and complex parts manufacturing. Since SLM uses a superimposed process, it also has some advantages in shortening the printing time. For example, NASA Marshall Space Flight Center (MSFC) launched the Additive Manufacturing Demonstrator Engine (AMDE) project in 2012, and designed an engine prototype that could be formed using SLM technology. The number of parts of the engine was reduced by 80%, the number of welds reduced from more than 100 to less than 30, and the development cycle reduced from 7 years to 3 years [3,4,5]. In the field of civil aerospace, the first flight test of the rocket attitude control power system cylinder assembly manufactured by Cloud Casting 3D using SLM technology was successfully completed. Compared with the original manufacturing technology, SLM formed gas cylinder components without any connecting tubes, reducing the weight of the product by 34.38%. It showed good impact performance, significantly reduced the structural dimensions, and greatly improved system reliability [6].

SLM is a technology that uses the principle of high-temperature melting and low-temperature solid forming of metal, a high-strength laser beam to melt metal powder, which finally cools and forms. SLM printing usually adds support to parts before printing preparation. The functions of supports are as follows:

(1) Prevent the part from collapsing due to excessive laser scanning;

(2) Restrain the warping problem caused by the cooling of parts, and at the same time, the parts can be related to maintaining the stress balance of the molding.

The advantages and disadvantages of the five main metal 3D printing technologies are compared, as shown in Table 1.

**Figure 1 micromachines-14-00057-f001:**
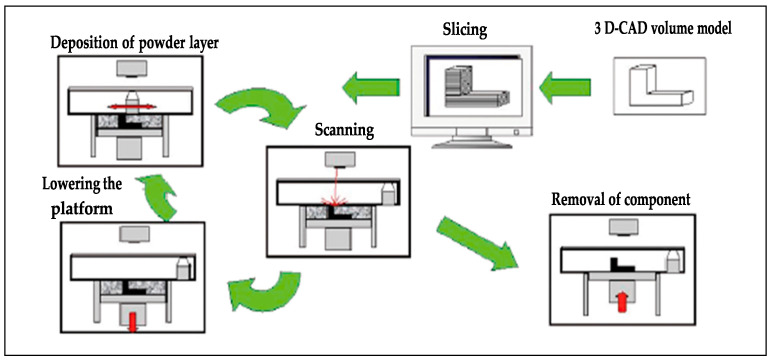
Principle of the SLM process [7].

## 2. Factors Affecting SLM Manufacturing

For manufacturing components with unique and complex geometric shapes, many factors must be considered to achieve the purpose of the industrial production of parts. Next, the related factors such as technology, design, and materials will be reviewed.

### 2.1. Process Parameters of SLM

The process parameters that affect the performance of SLM products mainly include the laser parameters, scanning strategy, powder spreading method, and support design, which have been studied by scholars all over the world. Yadroitsev et al. [8] analyzed the parameters of the SLM process. They pointed out that the process parameters include laser-related parameters, such as laser scanning power, scanning speed, pattern filling distance, scanning strategy, and laser spot size, as well as parameters related to construction, such as layer thickness, construction direction, and molding. In 2012, Yadroitsev et al. [9] conducted experiments on the influence of various parameters of 904L stainless steel monorail structure. Finally, ANOVA statistical methods were used to analyze the data of this experiment, and it was concluded that laser power and other parameters had a significant influence on SLM. Zhang et al. [10] studied the effect of laser power and scanning speed on SLM parts, and the results showed that both of them had an impact on the performance of the parts, and also on the density of the details.

#### 2.1.1. Laser Parameters

The laser is the core device in the SLM printing system, and it is the key device to making metal powder melt completely. Therefore, it is vital to choose a laser that is suitable for the SLM printing system. There are many kinds of lasers. Lasers can be divided into the solid lasers, gas lasers, liquid lasers, semiconductor lasers, chemical lasers, and excimer lasers [11]. Generally, the lasers needed for selective laser melting are mainly YAG lasers, CO2 lasers, fiber lasers, etc. Some scholars have conducted in-depth research on the influence of laser power on SLM, and found that among all the influencing factors, laser power has the most important influence on the parts, which is mainly that the energy of powder raw materials melting all comes from the laser equipment [12,13]. Therefore, Attar et al. [14] studied the density of commercially pure titanium parts manufactured with SLM by changing the laser power, and found that the pieces with a relative density higher than 99.5% can be realized by the ideal.

#### 2.1.2. Scanning Strategy

The existing research shows that the scanning strategy, including the length, direction, and distribution of scanning lines, will affect the temperature gradient when forming parts. Changing the scanning strategy can effectively reduce the residual stress and improve the quality of parts.

Currently, the common scanning paths include contour offset scanning, partition scanning, spiral line-filling scanning, scanning line-filling scanning, etc. [15]. The research shows that particular scanning strategies can reduce the temperature gradient and warpage [16,17]. The leading scanning methods are introduced here.

Profile scanning [18,19] mainly generates scanning paths by shifting inward along the boundary of each slice layer. The scanning strategy used is that the direction of scanning lines is not single but constantly changing, and the focus of internal stress is divergent. Following the heat transfer law, the residual stress can be effectively reduced. Huang [20] studied the warpage of parts in additive manufacturing through finite element simulation and experimental verification, and found that the warpage was proportional to the length of the scanning line. Jin et al. [21] studied the path planning of thin-walled structures in additive manufacturing, put forward an algorithm of generating wave path, and compared it with common ways, analyzing the advantages of this scanning path in molding quality and efficiency. The profile scan is shown in Figure 2.

The partition strategy can effectively reduce residual stress and improve molding efficiency. Chen et al. [22] studied the influence of scanning strategy on the temperature field of selective laser melting parts using finite element simulation. Under the zonal scanning mode, there would be remelting at the edge, and the overall temperature field distribution would be more uniform, which is beneficial for reducing residual stress. Zhao Yi et al. [23] studied the scanning methods of arbitrary contour polygons, and proposed a partition scanning algorithm based on searching extreme vertices. Zhao et al. [24] found advantages to the partition scanning strategy of the Fermat spiral line, which divides the scanning area into a group of sub-areas and scans the Fermat spiral line. This strategy could realize efficient and high-quality layered manufacturing. Ding et al. [25] proposed a partition scanning algorithm, which geometrically decomposed the area to be scanned into a group of convex polygons. For each convex polygon, a combination of zigzag scanning and contour offset scanning was used to determine the best scanning path. The partition is shown in Figure 3.

Spiral filling is a scanning line-filling method that follows the law of heat transfer in the forming process. The scanning line generates a scanning path spirally from the geometric center of the processing area to the periphery; hence, it is called the spiral filling method. The scanning path generated by the helix filling algorithm can avoid the problem of a single scanning direction of the scanning line, weaken the internal residual stress caused during the process of temperature reduction, and thus improve the physical properties of machined parts. The schematic diagram of helix-filling scanning is shown in Figure 4.

Scan line-filling is an algorithm similar to polygon area-filling in computer graphics [26]. The basic idea of this algorithm is to scan a polygon composed of several end-to-end line segments with horizontal scanning lines, from top to bottom (or from bottom to top), and each scanning line generates a series of intersections with some edges of the polygon. These intersections are sorted according to X/Y coordinates, and the sorted points are paired as two endpoints of the line segment, that is, two endpoints of the scanning line [27]. The range of scanning lines is determined by the maximum and minimum X/Y values of the vertices of a polygon. When the scanning line is calculated to the maximum or minimum X/Y value, the scanning line of the whole polygon is finished, and a series of corresponding point coordinates are obtained.

#### 2.1.3. Powder Spreading Method

SLM 3D printers use a scraper device to evenly spread the metal powder beaten in the powder bin onto the substrate or the upper layer of the processed plane, so the particle size of the powder affects the smoothness and thickness of the powder layer [28]. The uniformity of powder particle size is indispensable for SLM molding [29]. Therefore, the parameters related to powder, such as particle size distribution, shape regularity, fluidity, etc., have a significant influence on the molding of parts. Before processing, good planning and control are necessary.

#### 2.1.4. Support Design

The lightweight design of modern design parts mostly removes the material inside the part body, but this will result in many empty shells, especially for SLM technology. In the process of machining, due to the suspended structure inside the part, it is easy to cause great deformation of the region, thus affecting the forming of the region, and even causing manufacturing failure of the region. The supporting structure [30] mainly deals with the fact that detailed production cannot be finished because of excessively lightweight design. This structure can provide a stable supportive environment for the cantilever structure inside the elements, prevent the deformation of the elements in a limited way, keep the internal stress and increase the stability of the elements in the printing process.

Wang [31] put forward a new supporting structure called the “skin frame”. A new frame structure is used to directly replace redundant parts. Using this structure, the light weight of the details can reach 70%. Chen [32] proposed a truss-based internal support structure, which can directly put the microstructure into the printed workpiece as support, thus completing the design of the mount; it can be optimized now from the inside of the part, thus establishing the support. Yamanaka [33] proposed a unique support structure. This method first optimizes the part model to achieve specific quality characteristics, then builds a truss structure to process the inside of the part iteratively, and finally reaches the amount that meets the requirements. Hussein et al. [34] studied using SLM-based honeycomb structures for supporting metal parts. According to the conclusion of their experiment, two different lattice structures can be obtained (diamond and rotator) to reduce materials and construction time while meeting structural requirements. However, parts cannot be manufactured by the SLM process because of the vulnerability of volume densities that are too small [35]. Strano et al. [36] studied a hierarchical cell support structure in which more muscular cells were placed in the protruding part used elsewhere in the metal AM under the weight with less support. Gan [30] discussed “Y”, “IY”, and pin support structures based on finite element analysis to study and design the manufacture of thin plates and cuboids for SLM.

The process parameters of SLM technology play a qualitative role in the performance of parts after forming. Through the research of scholars in various countries, SLM technology is maturing.

### 2.2. Material Correlation Factors

#### 2.2.1. Absorption of Laser by Materials

In laser heat treatment, metal material is the central processing object, and its laser absorptivity is particularly important. According to Fresnel’s formula, the electric field of a light wave on the surface of a metal conductor constantly forms a standing wave node, and the free electron is forced to vibrate by the electromagnetic field of the light wave to produce secondary waves. These secondary waves cause strong reflected waves, which reflect most of the laser light. Especially in the long wavelength band, the photon energy is low, which can only act on the free electrons in the metal. It is almost totally reflected, with only a small amount of absorption. However, this small amount of absorption is significant in laser heat treatment. When the laser irradiates the surface of metal materials, most of the laser light is initially reflected due to excessive free electrons in the metal, and only a tiny part of the laser light is absorbed by the metal through the surface. On the other hand, while most of the laser light is reflected by the free electrons, a small part is absorbed by the bound electrons, excitons, lattice vibrations, and other oscillators in the metal. The reflection principle diagram of the material is shown in Figure 5.

The Fresnel formula is as follows:

Reflection principle:(1)$$\frac{A_{s1}^{\prime }}{A_{s1}}=\frac{n_{1}\cos i_{1}-n_{2}\cos i_{2}}{n_{1}\cos i_{1}+n_{2}\cos i_{2}}=-\frac{\sin\left(i_{1}-i_{2}\right)}{\sin\left(i_{1}+i_{2}\right)}$$
(2)$$\frac{A_{p1}^{\prime }}{A_{p1}}=\frac{n_{2}\cos i_{1}-n_{1}\cos i_{2}}{n_{2}\cos i_{1}+n_{1}\cos i_{2}}=-\frac{\mathrm{tg}\left(i_{1}-i_{2}\right)}{\mathrm{tg}\left(i_{1}+i_{2}\right)}$$

Refraction principle:(3)$$\frac{A_{s2}}{A_{s1}}=\frac{2n_{1}\cos i_{1}}{n_{1}\cos i_{1}+n_{2}\cos i_{2}}=\frac{2\sin i_{2}\cos i_{1}}{\sin\left(i_{1}+i_{2}\right)}$$
(4)$$\frac{A_{p2}}{A_{p1}}=\frac{2n_{1}\cos i_{1}}{n_{2}\cos i_{1}+n_{1}\cos i_{2}}=\frac{2\sin i_{2}\cos i_{1}}{\sin\left(i_{1}+i_{2}\right)\cos\left(i_{1}-i_{2}\right)}$$

Jiayao Zhang et al. [37] verified the relationship between powder particle size and laser absorptivity, and found that they were negatively correlated. As the particle size increased, fewer powder particles were irradiated, resulting in a more uneven irradiance and a lower intensity value. Similarly, Zhang D et al. [38] studied the different porosity and average of the powder layers. However, when the porosity decreased, a large amount of reflected light reduced the laser absorption rate of metal particles considerably. Still, this phenomenon will change with a decrease in powder particle size. This is because the reflection of the laser between spherical powder particles increases, which makes the absorption rate increase.

#### 2.2.2. Particle Size Distribution of Materials

The product quality of selective laser melting (SLM) is closely related to the powder particle size distribution. The small void ratio of the bed, mixed with coarse and fine powders, is beneficial to improve the density of SLM products. However, the fine-grained powder will lead to poor uniformity of powder distribution in SLM equipment, resulting in holes in SLM products.

Irrinki et al. [39] found that some parameters of metal powder and the melting process of metal powder affected SLM parts. At the same time, Attar et al. [40] studied the influence of particle size of powder on parts made with SLM. Relative densities of the samples produced by spherical particles were compared with those produced by irregular particles with a relative density of about 95%. On the contrary, powder characteristics are external parameters, because they are usually set by the supplying manufacturer or powder manufacturing supplier. In addition, SLM machine parameters are the same as laser types, and the maximum laser power and laser wavelength are related to the machine, and are limited in improving the performance of finished SLM parts. Therefore, it is possible to improve the performance of SLM products by optimizing process parameters to obtain completely dense components. Wilkes et al. [41] used research to show that nonmetallic ceramic parts manufactured by the SLM process would also be affected by the particle size and shape of the powder. Ng et al. [42] found that the grain size of laser-melted magnesium had a vast influence on the microstructure of SLM samples; with increased energy density, the grains in the melting zone would become coarser and coarser.

Du et al. [43], by studying the particle size distribution of 316L powder, found that an increase in acceptable powder amount (Figure 6) had a great influence on the surface morphology of SLM parts; the size and area of cracks became more prominent, and the relative density of the obtained product decreased (Figure 7).

From the microscopic point of view of materials, the laser absorptivity and particle size distribution of different materials are different. The main reason is that the types of raw materials that make up the materials are different. During SLM processing, differences in laser absorptivity and particle size distribution lead to differences in the processed parts. Research shows that the higher the laser absorptivity of materials, the finer the particle size distribution, and the better the SLM processing results.

## 3. Metal Materials

Although the development of SLM has gradually accelerated and is becoming commercialized, there are still few available metal materials [44,45]. Next, several common metal materials used in SLM equipment will be introduced, such as aluminum alloy, magnesium alloy, titanium alloy, and stainless steel.

### 3.1. Aluminum Alloy

Compared with many other SLM candidate materials, the treatment of aluminum alloy is more complicated [46]. This is mainly due to the lightweight nature and high fluidity of aluminum, which shows a high reflectivity wavelength range for typical SLM lasers, and has high thermal conductivity [47]. The absorption of aluminum by laser is poor; this indicates that it is necessary to quickly dissipate the heat of parts after high laser power processing [48]. Aluminum powder is very sensitive to oxidation [49]. Therefore, when printing aluminum alloy, attention must be paid to the oxygen content to avoid oxidation by oxygen in the air, and passivation [50]. Aluminum alloy is also the most widely used. It can be found in daily life and other fields [51]. The SLM aerospace aluminum alloy specimen is shown in Figure 8, The performance comparison of common aluminum alloy materials is shown in Table 2.

Zhao Xiaoming et al. [52] studied the relationship between the structure and properties of SLM Al Si10Mg aluminum alloy. They found that the grain, internal design, and mechanical properties of SLM AlSi10Mg alloy are superior to those of traditional cast AlSi10Mg parts. Buchbinder et al. [53] studied the influence of speed and scanning spacing on the hardness of AlSi10Mg. The research showed that the hardness of AlSi10Mg increased with an increase in scanning speed, and the best hardness was obtained when the laser scanning speed was 2500 mm/s. At the same time, the influence of scanning distance on the hardness of Al Si10Mg parts was determined. The results showed that the hardness of the samples obtained was independent of the scanning distance range used, and the optimal value was reached when the scanning distance was 0.15 mm. Currently, the optimal hardness value of AlSi10Mg parts processed by selective laser melting is twice that of die-cast AlSi10Mg parts. Rahman Rashet al. [54] used selective laser melting technology to print AlSi12 alloy powder and obtain circular, triangular, and hexagonal lattice structures. The experimental results showed that the bending strength of the triangular frame was (175.80 ± 1) MPa, and the bending strength of the circular and hexagonal lattice powders were (151.35 ± 0.67) MPa and (143.16 ± 3.85) MPa, respectively. It can be seen that the triangular structure has better bending resistance than other reinforced members. Karg et al. [55] studied the mechanical properties of 2219 aluminum alloy formed by selective laser melting. The results showed that the ultimate tensile strength of high-strength aluminum alloy after heat treatment is 384 MPa, and the elongation is 23%. Therefore, high-strength aluminum alloy formed by selective laser melting has good mechanical properties. Zhang [56] and others successfully prepared an almost entirely dense Al-4.24Cu-1.97Mg-0.56M cubic sample using selective laser melting technology. Research shows that the selection has good mechanical properties. Dynin [57] developed an advanced aluminum alloy, Al-10Si-0, 9Cu-0, 7Mg-0, 3Zr-0, 3Ce, for selective laser melting. The material showed a good speed build level in the SLM process, which allows for the formation of fine structures with low porosity levels. The microstructure characteristics of the alloy in the cast and T6 states were studied. The mechanical properties of Al Si Mg - (Cu) alloys produced by SLM exceeded those of such alloys produced by casting, which broadens their potential applications in aircraft, mechanical engineering, and other industrial fields. Hu et al. [58] systematically studied the change rule of surface roughness of AlCu5MnCdVA aluminum alloy during SLM, and drew the following conclusions: during SLM, the top surface roughness experienced a complex evolution; for single channel samples, the ray first decreased and then increased with an increase in scanning speed; and for single layer samples, the ray increased with an increase in scanning speed and shadow spacing. In addition, this value was lower than that of single-track samples manufactured at the same scanning speed. With an increase in the number of layers, the surface roughness first increases, then decreases, and finally tends stabilize. An increase in fluid flow intensity and actual powder thickness affects the roughness in both directions on the contrary, leading to the evolution of surface roughness. This evolution shows that the surface quality improves itself during processing, but the ability of self-improvement is limited. Once the surface quality is too poor and the process is unstable, it is difficult for the process to improve itself.

Some scholars studied the oxidation phenomenon of aluminum alloy powder in the SLM process. In the SLM process of aluminum and its alloys, a large number of an oxide is combined into parts so that it will form oxide film, as in the traditional way. Due to the active chemical properties of aluminum, a kind of passivation and reduction in the surface tension of parts is generated on the surface [59].

The low density, high strength, sufficient hardenability, good corrosion resistance, and excellent weldability of aluminum alloy make it suitable for a series of manufacturing industries, such as for automobiles, national defense, and aerospace equipment manufacturing [60,61].

### 3.2. Magnesium Alloy

Magnesium alloys are mainly used to develop lightweight structures because of their low densities. In addition, due to their biocompatibility, they offer the potential to be used as bioabsorbable materials for biodegradable bone substitute implants.

To achieve this goal, the DFG German Research Foundation, a research project funded by Deutsche Bank in 2012, used skull replacement with SLM technology to manufacture biodegradable implants for people. Its leading alloy implant is a biodegradable magnesium alloy scaffold, which is pre-activated with human cells of patients to support bone growth on implants before implantation. Due to the biocompatibility of magnesium metal in the human body, when the magnesium stent is implanted in the human body, the stent will gradually become compatible with the human body over time; at the same time, the injured bone tissue will also close.

In 2010, Ng et al. [62] studied selective laser melting magnesium to produce bone substitute scaffolding. Using the micro SLM system built by Hong Kong Polytechnic University, the experimental study was carried out in a protective gas environment. For the first test, coarse particles of 75–150 nm and fine spherical particles of 5–45 nm were produced, and satisfactory results were obtained. 

Krause et al. [63] conducted experiments on magnesium alloys. Their research showed that aluminum alloys with a calcium content of 8 ‰ had good human compatibility. Further research showed that the initial mechanical strength of magnesium alloys is insufficient when magnesium alloy stents are implanted into human bone joints. When low mechanical stability is necessary, it is appropriate to use these stents as the material of implants mixed with implants. WZ21 magnesium alloy contains biocompatible alloy, while rare earth elements show good biocompatibility, degrade slowly and homogenously in miniature abdominal cavities, and were shown to have stable mechanical integrity in the femur of experimental pigs and rats [64,65]. WE43 alloy is a superb alloy for manufacturing biodegradable stents. Therefore, this material is often used as a magnesium alloy implant to treat patients. Yuan et al. [66,67] used additive manufacturing technology to optimize the structure of MgNdZnZr magnesium alloy, and designed and prepared three magnesium scaffolds with the same porosity and average pore size using SLM technology. It was found that the SLM magnesium alloy stent presented a fully connected structure, appropriate compression performance, and moderate degradation behavior, indicating that the stent printed by SLM has good clinical application prospects.

Magnesium alloy biodegradable implants will be applied in the future of bone sutures.

In recent years, because of its good bearing capacity, magnesium alloys have been the preferred material for biodegradable polymers. SLM technology alloys of magnesium can be used to manufacture single biodegradable implants with complex shapes. Generally speaking, magnesium and its alloys are quite mature in SLM applications.

### 3.3. Titanium Alloy

Compared with other materials that can be used for SLM printing, titanium alloy is the ideal printing material because of its cleanness and high utilization rate in processing, and titanium can be used to print parts with complex shapes on the SLM platform.

Titanium alloy is widely used in biomedical implants [68], among which Ti-45Nb is the primary material of implants, mainly due to its low Young’s modulus (~62Gpa) and high cooling rate. In addition, Ti-45Nb has excellent elasticity, good corrosion resistance, and good chemical resistance, mainly due to the passivation layer formed on the surface of titanium components [69].

Huang et al. [70] used SLM technology to conduct heat treatment research on TA15 titanium alloy style. The research shows that the microstructure of TA15 titanium alloy style after heat treatment is a typical basket structure, and the toughness of parts can be improved through heat treatment. Finally, with an increase in heat treatment time, the dimple size of the fracture increases. The failure mode changes from brittle fracture to semi ductile fracture, and finally to complete ductile fracture.

These results show that SLM technology is different from other manufacturing methods, and can make full use of the potential of solid titanium. In addition, porous titanium structure with the necessary characteristics can still be manufactured using SLM while retaining its biomechanical properties [71]. However, the reliability of biomedical application parts of titanium alloy produced by SLM, especially its powder materials, its corrosion behavior, and the influence of fatigue properties, still needs further study. SLM parts of titanium alloy are shown in Figure 9.

### 3.4. Stainless Steel

316L stainless steel, also known as titanium steel, is characterized by solid point corrosion resistance, excellent high-temperature strength, and non-magnetism. It is widely used in pipelines, the food industry, in marine and biomedical equipment, fuel cells, and other fields [72,73,74]. In recent years, 316L stainless steel has gradually become the raw material of SLM 3D printing technology, and has been studied by many scholars [75]. SLM bridge model of 316L stainless steel is shown in Figure 10.

Kruth et al. [76] conducted many novel types of research on the SLM manufacturing of stainless steel. Soon after, Kruth [77] proved that changes in process parameters play a vital role in SLM for forming parts. Since then, many scholars have researched this area and reached similar conclusions [78,79,80]. However, the study showed that it is difficult to achieve the entire density state because there is a relatively fixed porosity in the metal powder. On the other hand, lack of a particular mechanical pressure will also make the total density unable to reach its highest [81]. Casati et al. [82] reported that a reduction in tensile strength would make the elongation break, attributing very different power of SLM 316L stainless steel, mainly because of partially melted powder particles in their microstructure. To solve this problem, laser remelting is the method applied; that is, before spraying the next layer, the powder layer is scanned twice. Yasa et al. [83] proved that the density of stainless steel parts could be improved, and the surface roughness could be reduced by the secondary scanning method, thus improving the fatigue characteristics. Riemer et al. [84] found that 316L stainless steel printed with SLM technology showed anisotropy in notched high-cycle fatigue behavior and fatigue crack. The research of Zhou Yuecheng et al. [85] on 316L stainless steel showed that SLM is suitable for manufacturing small- and medium-sized parts, and that the mechanical properties of SLM products show obvious anisotropy and inhomogeneity due to epitaxial growth crystals, micro defects, weld pool boundaries, and residual stresses in the microstructure.

The research on 316L stainless steel by scholars from all over the world using SLM technology shows that due to certain porosity between stainless steel powders, the density state of stainless steel cannot be guaranteed, which can cause internal defects in the parts printed with 316L stainless steel, affecting the overall service life of the parts, and even leading to parts being scrapped. At the same time, scholars are also using methods to constantly improve the SLM process of stainless steel; the research on stainless steel is also maturing.

To sum up, the SLM process has shown strong developments in the printing of various metal materials. Although the material and application performance of parts still requires further exploration and verification, and the manufacturing process still needs to be improved, SLM metal printing technology has shown great application prospects in various fields: in the medical field, for patient specific implants and other high-value medical device components; in the field of automobile manufacturing, high-speed prototype design and customized components, or low volume, high-value applications; in the field of aerospace, conduits and other components. SLM metal printing is an exciting technology with many potential applications. With continued growth in uses and the maturity of technology, the process and materials are becoming cheaper. We should see SLM metal printing becoming increasingly common.

## 4. Nonmetallic Materials

As an inorganic and nonmetallic material, ceramics are widely used in modern industries such as automobiles, aerospace, national defense, and machining. This mainly depends on the low thermal conductivity and high wear resistance of ceramic materials [86,87,88,89]. Due to the biocompatibility of ceramics, ceramics have also been studied in the medical field, among which ceramic teeth are the most widely used, in addition to uses in other prostheses, stents, etc. Several ceramic parts printed by SLM technology are shown in Figure 11.

Zirconia is an excellent ceramic material, mainly due to its high mechanical properties, compression resistance, tensile resistance, and excellent corrosion resistance [91]. Due to its corrosion resistance and excellent physical properties, zirconia can be used in the field of full-mouth porcelain teeth in medicine [92]. Zheng et al. [93] researched and found that when SLM was used to treat ceramic paste for printing, cooling cracks occurred in the curing process of parts due to thermal expansion and cold contraction. Shishkovsky et al. [94] found similar results when studying the SLM of yttria-stabilized zirconia powder with aluminum or aluminum oxide in different oxygen and argon gas streams. Thirdly, due to the existence of macropores larger than 100 μm and the appearance of cracks, the mechanical strength decreases. In addition, it was reported that adding a small amount of zirconia made with SLM to high-temperature nickel alloys can enhance the mechanical properties of nickel alloys [95].

Silicon carbide particles are a common reinforcing material, often used in aerospace and military equipment because of their high strength, high hardness, high temperature resistance, and other advantages. At the same time, they are cheap and easy to use [96]. It was found that in the SLM forming process, under good process parameters, a good metallurgical combination can occur between 10% SiC particles and Al Si10Mg to generate a new phase Al4SiC4, with a refined grain size and improved composite properties. Further research shows that the strength of 10% SiC reinforced AlSi10Mg composite specimens formed with SLM is greatly improved compared with the conventional casting process; the materials can achieve high density and strength under the appropriate combination of process parameters [97,98].

The best combination of the metal matrix of reinforced ceramic matrix composites and more complex and stronger ceramic reinforcement materials improves wear resistance, strength, and high-temperature mechanical properties [99].

In situ ceramic composites show more advantages. Compared with pre-added composites, the wettability between ceramic and metal is better, mainly because the molecular distribution of compounds is relatively uniform [100].

However, there are new challenges for ceramics in SLM:

1. It is necessary to develop a laser source that is capable of releasing wavelengths that can be absorbed by ceramics.

2. Ceramic parts will produce more pores due to their high viscosity.

3. Ceramic parts will crack at high temperatures, so stress research is needed.

## 5. Conclusions

In this paper, the research progress of selective laser melting technology was reviewed. The influences of SLM process parameters on forming defects and mechanical properties were summarized in detail. The properties of magnesium, aluminum, titanium alloy, and stainless steel made with SLM were also summarized. Currently, the mechanical properties of SLM manufacturing alloys are obviously superior to those of casting alloys. Due to the characteristics of SLM additive, it has great advantages in preparing large and complex components. Therefore, SLM manufacturing of components has great engineering application potential. According to current research and development trends at home and abroad, future SLM research needs to focus on the following aspects:

(1) The existing equipment has low feed speed, unstable work environments, and limited processing size. Therefore, the stability of equipment processing should be improved, feed speed and processing size should be increased, work efficiency should be improved, and parts with uniform organization and good performance should be manufactured.

(2) The comprehensive properties of powders require improvement, such as powder particle size, laser melting mechanism, thermophysical properties, etc., and a suitable production process must be found to solve the defects of spheroidizing effect, warping deformation, and cracking of machined parts.

(3) Complete industrial chains for engineering application should be developed. Break through the shackles of traditional manufacturing technology, and form an entire industrial chain, such as a design mode, production mode, detection means, processing, assembly, etc., in order to meet the ever-developing demands of new manufacturing technologies.

SLM technology can process dense parts with complex shapes, good surface roughness, and high dimensional accuracy; thus, SLM can be used to process high-temperature alloys, and other factors. The processing technology is simple, which provides faster methods for product design and production. SLM technology can also be applied to rapidly manufacture conceptual prototypes, manufacturing molds, functional parts, etc. [101,102]. Therefore, SLM technology is a significant development direction of additive manufacturing technology. Currently, China needs to continue its efforts in SLM technology, to improve equipment, improve the comprehensive performance of powders, and form complete industrial chains to meet the needs of new manufacturing technologies.

## Figures and Tables

**Figure 2 micromachines-14-00057-f002:**
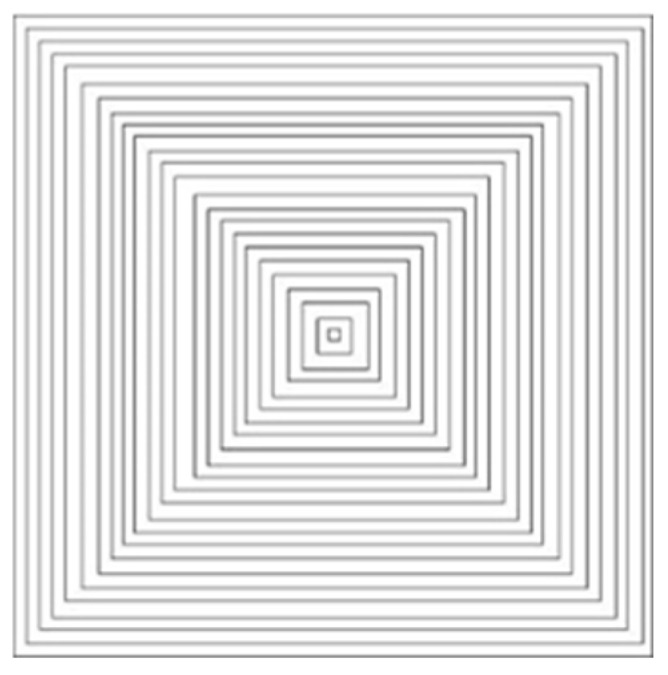
Schematic diagram of contour offset scanning.

**Figure 3 micromachines-14-00057-f003:**
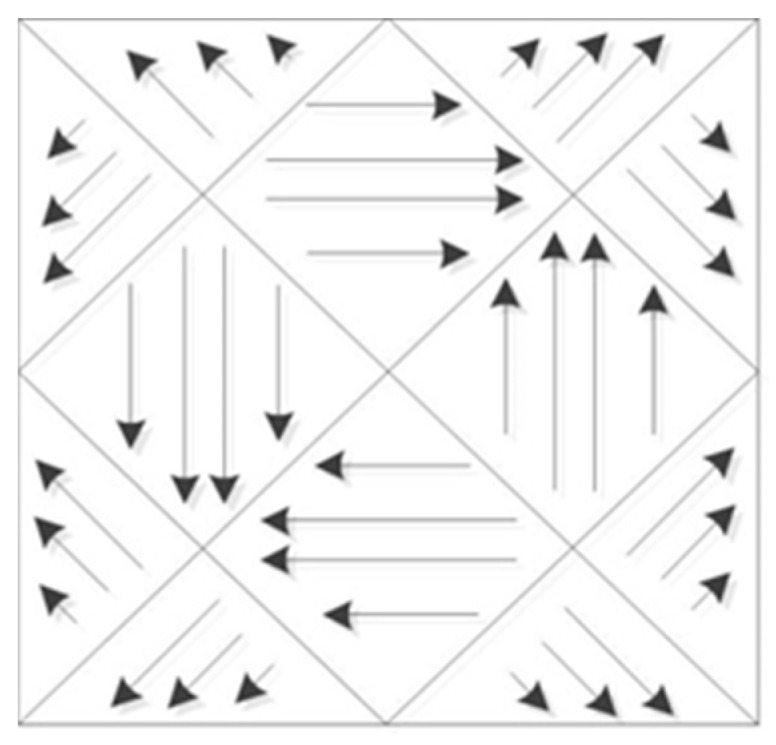
Schematic diagram of partition scanning.

**Figure 4 micromachines-14-00057-f004:**
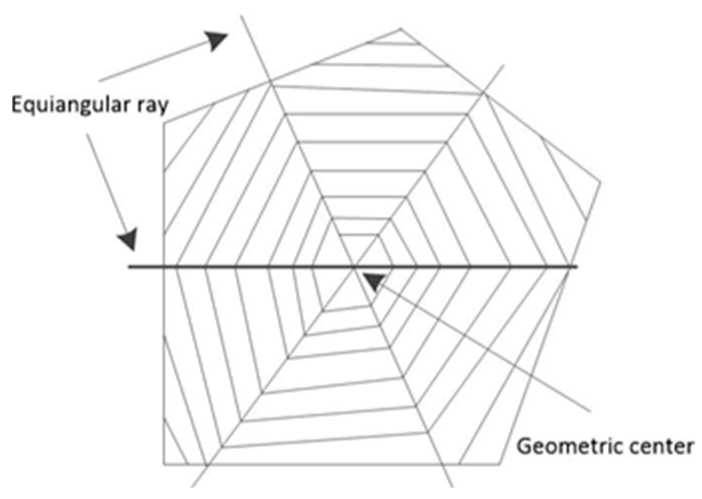
Schematic diagram of spiral fill scanning.

**Figure 5 micromachines-14-00057-f005:**
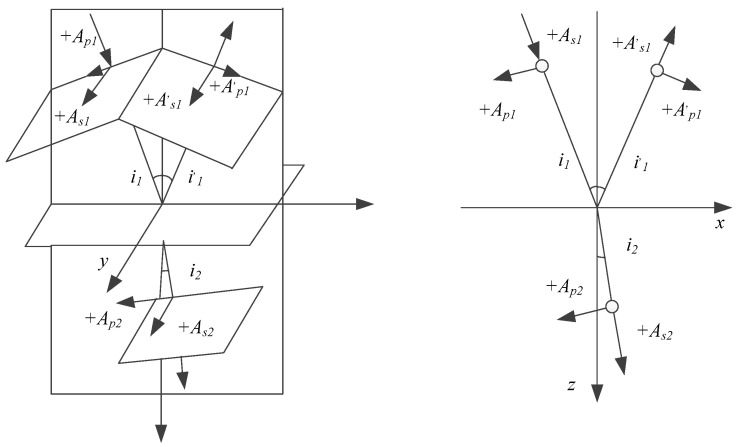
Reflection and its schematic diagram.

**Figure 6 micromachines-14-00057-f006:**
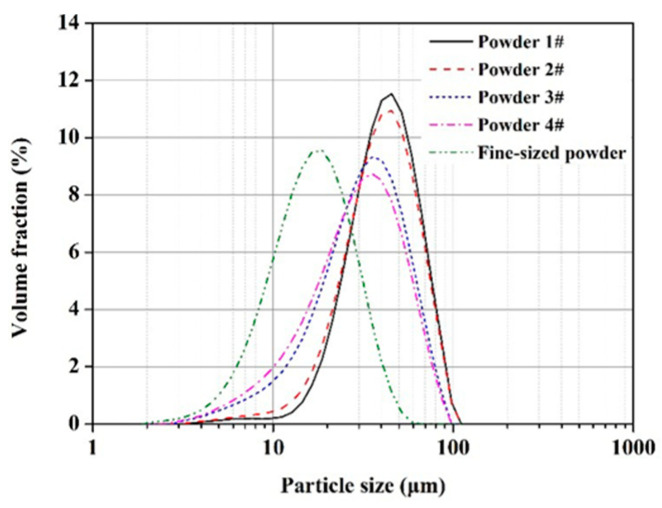
Particle size distribution of powder [43].

**Figure 7 micromachines-14-00057-f007:**
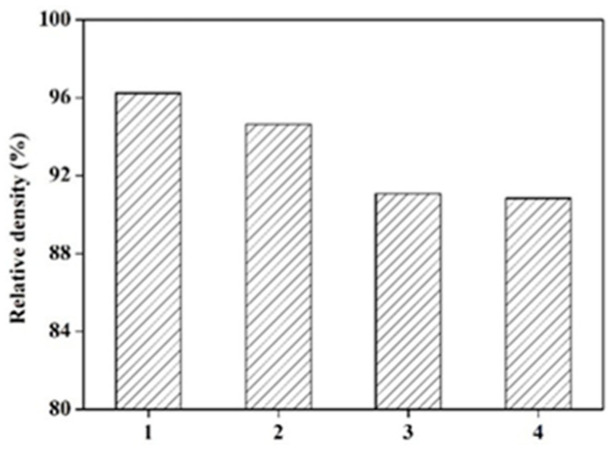
Reflection and its schematic diagram [43].

**Figure 8 micromachines-14-00057-f008:**
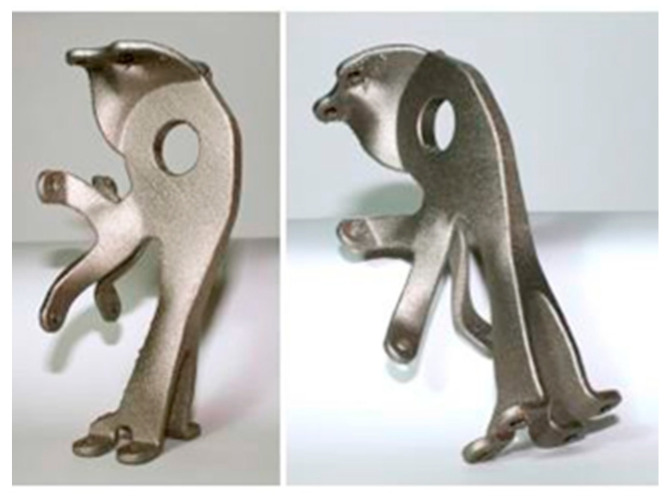
Aluminum alloy parts produced by SLM [51].

**Figure 9 micromachines-14-00057-f009:**
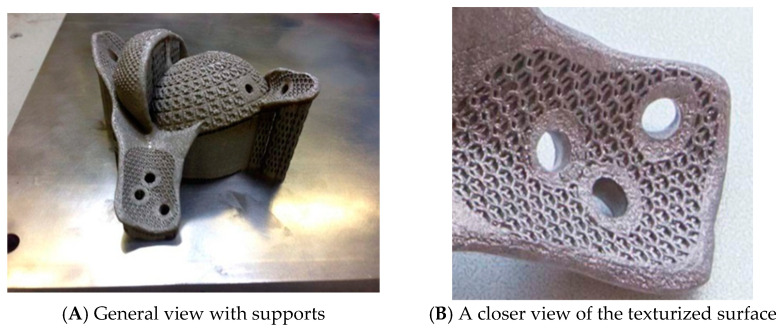
Titanium alloy hip implants produced by SLM [71].

**Figure 10 micromachines-14-00057-f010:**
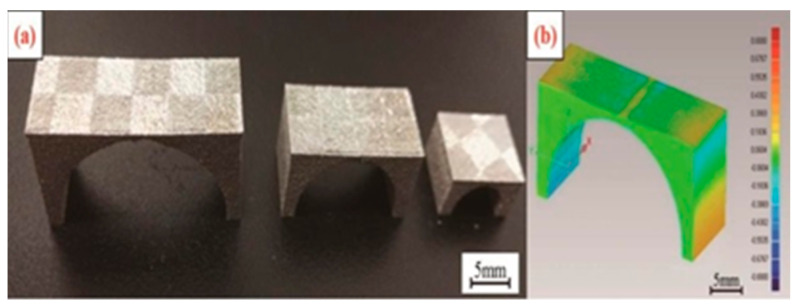
316L Bridge model made by SLM: (**a**) A sample of a bridge-shaped structure; (**b**) the deformation diagram [75].

**Figure 11 micromachines-14-00057-f011:**
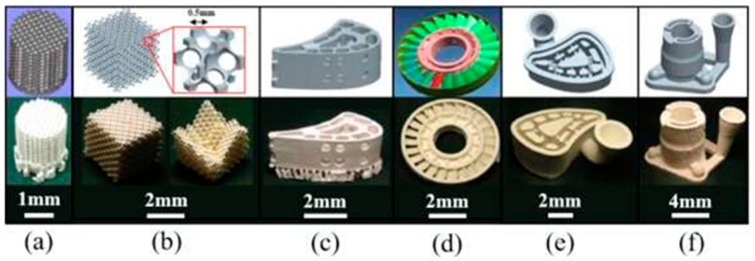
Advanced ceramic parts manufactured with SLM [90]: (**a**) porous bioceramic scaffolding; (**b**) photonic crystals; (**c**) hollow turbine blade; (**d**) impeller; (**e**,**f**) investment casting moulds.

**Table 1 micromachines-14-00057-t001:** Common metal 3D printing processes.

Process	Merits	Demerits
DMLS	1. Metal alloy or alloy powder 2. Unsintered or melted materials can be reused 3. Printing can achieve high precision	1. Expensive 2. The porosity of the print is high 3. Metal powder is sintered rather than melted
SLM	1. The density of processed standard metal exceeds 99% 2. Good mechanical properties are equivalent to the traditional process 3. Parts can be used for later welding	1. Poor resolution 2. Poor surface roughness, residual internal stress
EBM	1. Different kinds of metals with fusible high melting point 2. The resulting residual stress is very low 3. Customizable design, higher processing efficiency 4. It is suitable for the processing of medical devices with better biocompatibility	1. The cost of equipment is expensive 2. It causes environmental pollution 3. Non-conductive materials must be used
DED	1. Easy access to raw materials 2. Low processing cost	1. Lower resolution accuracy 2. Higher surface roughness 3. The complexity of parts may be limited 4. Requires post-processing

**Table 2 micromachines-14-00057-t002:** Grades and principal properties of aluminum alloy powder materials required for additive manufacturing.

Material	Tensile Strength/Mpa	Yield Strength/Mpa	Elongation/%
AlSi_10_Mg	460 ± 20.0	270 ± 10.0	9 ± 2%
AlSi_12_	409 ± 20.0	211 ± 20.0	5 ± 3%
AlSi_7_Mg	294 ± 17.0	147 ± 15.0	3%
AlSi_9_Cu_3_	415 ± 15.0	236 ± 8.0	5 ± 1%
AlCuMgZr	451 ± 3.6	446 ± 4.3	—

## Data Availability

Not applicable.

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
