# Peer review of "A Review of Research Progress in Selective Laser Melting (SLM)"

_micromachines, 2022, doi:10.3390/mi14010057_

Round 1
Reviewer 1 Report
The topic of this research is attractive and can be useful for a large number of researchers. Selective Laser Melting (SLM) is a particular rapid prototyping, 3D printing, or Additive Manufacturing (AM) technique designed to use high power-density laser to melt and fuse metallic powders. This technique has been proven to produce near net-shape parts up to 99.9% relative density. Recent developments of fibre optics and high-power laser have also enabled SLM to process different metallic materials, ceramic and composite materials.
The abstract and introduction provides a good, generalized background of the topic that quickly gives the reader an appreciation of the wide range of applications for this technology. However, to make the paper more substantial, the author may wish to provide several sentences to substantiate the claim made in the abstract.
After the statement about the importance of the SLM technique in the abstract the author offers no explanation of why he chooses the SLM in the present work. The table is shown, without further explanation...
The motive, goals and significance of the research should be stated more clearly, and to differentiate the paper some more from other applied papers
Provide several references to substantiate the claim made in the third and fourth sentence of the introduction (that is, provide references to other groups who do or have done research in this area).
The scientific paper is not clearly structured and methodologically laid out, so that the initial assumptions and expected deliverables are not clearly emphasized.
The conclusion is too general and not fully aligned with the research.
Author Response
Dear reviewer
First of all, thank you for your review of this article.
In the process of modification, we have made adjustments according to your suggestions, which will be explained for you next.
- After the statement about the importance of the SLM technique in the abstract the author offers no explanation of why he chooses the SLM in the present work. The table is shown, without further explanation...
The author's Reply: In the summary of the article, we added some content to explain why SLM was selected. In addition, in Table 1, we added content to make the comparison between the advantages and disadvantages of metal additive manufacturing more sufficient and deleted the FDM printing method because of the poor effect and poor use of metal printing in FDM.
- The motive, goals and significance of the research should be stated more clearly, and to differentiate the paper some more from other applied papers
The author's reply: The motivation and goal of the article can be reflected in the last part of the abstract, and the writing significance of the article is described in the final summary part of the article.
- Provide several references to substantiate the claim made in the third and fourth sentence of the introduction (that is, provide references to other groups who do or have done research in this area).
The author's reply: In the second paragraph of the article, we interpreted the relevant literature and cited the references.
- The scientific paper is not clearly structured and methodologically laid out, so that the initial assumptions and expected deliverables are not clearly emphasized.
The author's reply: We have rewritten the structure of the article to make it meet the requirements; In order to make the structure of the article more rigorous, we made a summary of this paragraph at the end of each part.
- The conclusion is too general and not fully aligned with the research.
The author's reply: We have made some changes to the conclusion, which can be echoed with the abstract of the article, and the conclusion can meet the requirements.
The modified part has been identified with yellow background
Finally, thank you again for your review of this article and your comments.
With best wishes
Hongjian Zhao

Reviewer 2 Report
See attached.

Author Response
Dear reviewer
First of all, thank you for your review of this article.
In the process of modification, we have made adjustments according to your suggestions, which will be explained for you next.
- What DMLS, BEM, DED and FDM are short for are not clarified in Table 1, nor in the text. Please.
The author's Reply: We followed your advice and added List of Abbreviations at the end of the article to explain abbreviations and deleted the FDM printing method because of the poor effect and poor use of metal printing in FDM.
- Fig. 5 is in low quality, and the critical parameters need to be clearly marked and easily identified by the reader.
The author's reply: Following your suggestion, we redraw the figure in Figure 5 in the form of vector diagram to ensure the clarity of the picture
- In Table 2, AlSi10Mg, AlSi12, AlSi7Mg and AlSi9Cu3 are not “Brand”, but “Materials”. Furthermore, the number should be subbed, such as AlSi10Mg.
The author's reply: We adopted your suggestion and have modified it.
- The unit of tensile and yield strengths should be MPa, not Mp.
The author's reply: We adopted your suggestion and have modified it.
- Fig. 3 and Fig. 4 are not easy for readers to understand and need to be explained and illustrated in detail in caption parts.
The author's reply: We explained in the picture title of Figure 3 and Figure 4
- The formulas in section of 2.2.1 needs to be numbered.
The author's reply: We have adopted your suggestions and have numbered the formulas here.
For others, the spelling and grammar errors that need to be corrected have been corrected
The modified part has been identified with yellow background
Finally, thank you again for your review of this article and your comments.
With best wishes
Hongjian Zhao

Reviewer 3 Report
This reviewer strongly suggests to reject the submission for the following reasons:
(a) The authors (e.g. the corresponding author) have published no single piece of paper in the field of selective laser melting additive manufacturing, at least not seen from the reference list of the current submission.
(b) The authors have tried to summarize research progress in the field of selective laser melting additive manufacturing, with a title of 'A review of recent progress in selective laser melting'. The references used by the authors, however, are rather old. Many of the reference papers were published more than 10 years ago. No papers published in 2020, 2021 and 2022 were ever referred to.
(c) People have already published thousands of papers on the topic of 'selective laser melting'. A review paper is expected to be ~50 pages rather than ~14 papers that of the current submission (it also has ~4 pages for reference list). The submission is too superficial.
Author Response
Dear reviewer
First of all, thank you for your review of this article.
Having seen your comments, I made a profound reflection on this article and revised its contents.
I will reply to your message:
- Although the author has not published a paper on selective laser melting technology, I want to say that when every scholar sets foot in a new research field, although the relevant articles have not been published, it does not mean that the author's ability is insufficient,
- I am very sorry for the issue of the reference year of the literature. We have rectified the literature, added the content of articles and references in the last three years, and made a corresponding summary.
- Although the content of the article may be insufficient, the content of the article is also very rich, although the sparrow has all the dirty things.
The above is the reply to you
Finally, thank you again for your review of this article and your comments.
The modified part has been identified with yellow background
With best wishes
Hongjian Zhao

Reviewer 4 Report
There are already several reviews of this type. The authors must highlight what is new in this review and include more recent publications. I feel the review is not wide enough for publication
Author Response
Dear reviewer
First of all, thank you for your review of this article.
Having seen your comments, I made a profound reflection on this article and revised its contents.
I will reply to your message:
Although there are many related papers, each one has its own characteristics.
In this paper, the latest development of selective laser melting technology is reviewed and compared with other metal 3D printing technologies. The necessary parameters of SLM technology, standard printing methods and printing materials are introduced in detail, and the influence rules of SLM process parameters on forming defects and mechanical properties are summarized in detail. The properties of magnesium aluminum titanium alloy and stainless steel manufactured by SLM are summarized. According to the problems and application prospects of SLM technology, the future development direction of SLM technology is pointed out.
The above is the reply to you.
Finally, thank you again for your review of this article and your comments.
The modified part has been identified with yellow background
With best wishes
Hongjian Zhao

Round 2
Reviewer 2 Report
accept
Reviewer 4 Report
NA